# *Agrobacterium tumefaciens*-Mediated Genetic Transformation of the Ect-endomycorrhizal Fungus *Terfezia boudieri*

**DOI:** 10.3390/genes11111293

**Published:** 2020-10-30

**Authors:** Lakkakula Satish, Madhu Kamle, Guy Keren, Chandrashekhar D. Patil, Galit Yehezkel, Ze’ev Barak, Varda Kagan-Zur, Ariel Kushmaro, Yaron Sitrit

**Affiliations:** 1The Albert Katz International School for Desert Studies, The Jacob Blaustein Institutes for Desert Research, Ben-Gurion University of the Negev, Beer Sheva 84105, Israel; lsatish@post.bgu.ac.il (L.S.); madhu.kamle18@gmail.com (M.K.); guyguy123@gmail.com (G.K.); cdshekhar.patil@gmail.com (C.D.P.); gality@bgu.ac.il (G.Y.); 2Avram and Stella Goldstein-Goren Department of Biotechnology Engineering and The Ilse Katz Center for Meso and Nanoscale Science and Technology, Ben-Gurion University of the Negev, Beer Sheva 84105, Israel; arielkus@bgu.ac.il; 3Department of Life Sciences, Ben-Gurion University of the Negev, Beer Sheva 84105, Israel; barakz@post.bgu.ac.il (Z.B.); zur@bgu.ac.il (V.K.-Z.)

**Keywords:** *Agrobacterium tumefaciens*, desert truffles, Ect-endomycorrhizal fungus, genetic transformation, hygromycin, *Terfezia boudieri*

## Abstract

Mycorrhizal desert truffles such as *Terfezia boudieri*, *Tirmania nivea*, and *Terfezia claveryi*, form mycorrhizal associations with plants of the Cistaceae family. These valued truffles are still collected from the wild and not cultivated under intensive farming due to the lack of basic knowledge about their biology at all levels. Recently, several genomes of desert truffles have been decoded, enabling researchers to attempt genetic manipulations to enable cultivation. To execute such manipulations, the development of molecular tools for genes transformation into truffles is needed. We developed an *Agrobacterium tumefaciens*-mediated genetic transformation system in *T. boudieri*. This system was optimized for the developmental stage of the mycelia explants, bacterial optical density, infection and co-cultivation durations, and concentrations of the selection antibiotics. The pFPL-Rh plasmid harboring *hph* gene conferring hygromycin resistance as a selection marker and the red fluorescent protein gene were used as visual reporters. The optimal conditions were incubation with 200 μM of acetosyringone, attaining a bacterial optical density of 0.3 OD_600_; transfer time of 45 min; and co-cultivation for 3 days. This is the first report on a transformation system for *T. boudieri*, and the proposed protocol can be adapted for the transformation of other important desert truffles as well as ectomycorrhizal species.

## 1. Introduction

Desert truffles are mycorrhizal fungi belonging to the Ascomycota phylum that produce hypogeous edible fruit bodies. The genera of *Terfezia* and *Tirmania* establish mycorrhizal associations with their host plants, mainly *Helianthemum* species, which reside in arid and semiarid ecosystems [1,2]. *Terfezia* species are proficient in forming various types of mycorrhizal lifestyles, including ectomycorrhiza, endomycorrhiza, and ect-endomycorrhiza, under defined conditions [2,3,4,5,6,7]. In natural habitats, the formation of mycorrhizal symbiosis is beneficial for plant survival in boreal forests, savannas, or deserts. In fact, most terrestrial plants harbor some type of mycorrhiza, such as arbuscular mycorrhiza, ericoid mycorrhiza, and ectomycorrhiza [8]. The host plant benefits from the mycorrhizal association that improves its water supply and the uptake of phosphate ions and other minerals. Indeed, mycorrhizal plants grown under arid conditions exhibit a better physiological performance compared to non-mycorrhizal plants—i.e., better water use efficiency, photosynthesis, respiration, and development [6]. *T. boudieri* forms a mycorrhizal association with its host plant *Helianthemum sessiliflorum*, a small perennial desert shrub, and with other species of the Cistaceae [6,9].

*Terfezia boudieri* (Pezizaceae), the most common desert truffle, is a socioeconomically important wild edible fungus that is consumed in countries around the Mediterranean basin up to the Persian Gulf and used both as a medicine and as a highly valued food in modern cuisine [4,10,11,12]. Desert truffles have a high content of minerals, amino acids, proteins, fibers, fatty acids, and carbohydrates. Moreover, they are rich in anti-bacterial, anti-cancer, and anti-oxidant compounds contributing to human health [4,6,13,14,15]. These features result in truffles providing an important nutritional resource, especially for nomadic tribes in desert areas where truffles serve as a staple seasonal food [16]. For these reasons, research has been carried out on desert truffle cultivation, biology, and genetics [17,18], and the development of these valued fungi as agricultural crops is in high demand [19,20]. Despite the potential to develop these valued fungi as crops, this has as yet not occurred.

Recently, the genome sequences of several desert truffles [21,22] have been completed. Despite this, the genomic information available for many of these species is very limited when compared to other mycorrhizal fungi [7]. Here, we established an experimental system of *T. boudieri* that forms mycorrhiza with *Cistus incanus* hairy roots [3] and adopts different lifestyles under different conditions [3]—i.e., ecto- vs. endomycorrhiza. Employing *T. boudieri* and *C. incanus* hairy roots as a model system offers the advantage of fast-growing mycelia and ever-growing roots, enabling us to study the physiology and regulatory genes involved in the switch between lifestyles. However, to meet these goals, it was first necessary to develop an amenable and stable transformation system.

The *Agrobacterium tumefaciens*-mediated transformation system is an effective tool for the functional genomic research of fungi [23]. So far, *A. tumefaciens* transformation into different tissues, such as hyphae, spores, sporocarp, protoplast, or gill fragments has been employed to study various fungal species, including Ascomycetes, Basidiomycetes, and Zygomycetes [24,25,26]. Reports on *A. tumefaciens* transformation in ectomycorrhizas using mycelia include *Tuber borchii* truffle [23,27]; *Hebeloma cylindrosporum*, *Suillus bovinus*, and *Paxillus involutus* [28]; *S. bovinus* [29]; *H. cylindrosporum* [25,30,31]; *Pisolithus microcarpus* [32]; *Laccaria bicolor* [33,34,35,36,37]; and *L. laccata* [38]. Several other conventional methods include the use of protoplasts in *H. cylindrosporum* [39], microprojectiles in *Gigaspora rosea* [40], and *Pisolithus tinctorius* [41]. Still, ectomycorrhizas could not have been effectively exploited for functional genomics studies as a consequence of the lack of stability in previous transformation attempts [23]. Brenna et al., in a detailed work, demonstrated that several parameters such as plasmid type and promoter choice are critical for a high transformation efficiency. They compared the conventional plasmid pBGgHg to pABr1 and pABr3. The transformation efficiency was increased six-fold when they employed pABr3 under the control of constitutive *ToxA* promoter compared to pBGgHg harboring *glyceraldehyde-3-phosphate dehydrogenase* promoter and *CaMV35S* terminator [23]. Moreover, due to their poor genetic tractability, including transformation, truffles have so far eluded in-depth genomic research [42]. Therefore, we focused on developing a much-needed transformation system for mycorrhizal desert truffles. The two forms of *T. boudieri* lifestyles, ecto- or endomycorrhiza, enable us to study the regulatory mechanism involved in the switch from one life style to another and manipulate the genes involved in this process.

Here, we describe the various optimized parameters for a successful *Agrobacterium*-mediated transformation of *T. boudieri*. To the best of our knowledge, this is the first report of a high-frequency stable *A. tumefaciens*-mediated genetic transformation in desert truffles.

## 2. Materials and Methods

### 2.1. Terfezia Boudieri Culture

*T. boudieri* strain 3-27-01 was grown on agar plates containing Fontana medium (pH 5.6 for two weeks, at 24 °C, in the dark) [43]. Prior to transformation, mycelial cylinders (0.75 cm^2^) were excised and transferred to induction plates. The cylinders were placed on Fontana induction medium supplemented with 100 µM of acetosyringone and topped with a Whatman No. 1 filter paper (9 cm diameter) for 3 days. With the filter paper, it was easier to further transfer the cylinders to selection plates.

### 2.2. Hygromycin and Geneticin Sensitivity Test

*T. boudieri* cultures were tested for their antibiotics sensitivity. Mycelia were grown on Fontana-agar plates supplemented with hygromycin at concentrations of 1.5, 2.5, and 5 mg/L and geneticin at 5, 10, and 15 mg/L. The culture plates containing the mycelial cylinders were grown at 24 °C in the dark for 2 weeks.

### 2.3. A. tumefaciens Culture and Plasmid Transfection/Electroporation

The plasmid pFPL-Rh contains hygromycin phosphotransferase (*hptII*) as a selection marker and the red fluorescent protein (RFP) as a reporter, while the plasmid pFPL-Gg contains geneticin as a selection marker and the green fluorescent protein (GFP) as a reporter. Competent *A. tumefaciens* AGL1 cells were grown and prepared for transfection as previously described [44]. Briefly, plasmids pFPL-Rh (11.5 kb) or pFPL-Gg (11.5 kb) (Addgene, Watertown, Massachusetts, USA) were transformed into competent cells using 0.2 cm electroporation cuvettes in a Micro-Pulser electroporator (Bio-Rad, Hercules, CA, USA) at 25 µF, 200 Ω, and 2.5 kV. Single colonies were isolated on selective YEP agar plates at 28 °C and supplemented with respective antibiotics [45]. Bacterial cultures were grown in 5 mL of YEP broth with antibiotics and plasmid DNA was extracted. Positive colonies were verified using gene-specific primers.

For transformation experiments, a freshly grown single colony of *A. tumefaciens* harboring *pFPL-Rh* or *pFPL-Gg* plasmid was suspended in 5 mL of YEP liquid medium with the appropriate selection antibiotics [45]. The culture was incubated for 12 h at 28 °C with shaking (220 rpm) on an orbital shaker, and 5 µL of this culture was suspended in 50 mL of YEP liquid medium and incubated at the same conditions. When the culture reached an optical density of 0.6 (OD_600_), it was pelleted by centrifugation at 4000 rpm for 20 min at 28 °C. The resulting pellet was re-suspended in 30 mL of Fontana liquid medium, re-incubated under the same conditions for 2 h, and diluted to different optical densities for further experiments.

### 2.4. T. boudieri Infection and Co-Cultivation with A. tumefaciens

In optimization experiments, the mycelium developmental stage (according to the peripheral, middle, and interior regions of mycelia grown on agar plate) and effective concentrations of acetosyringone (100 and 200 µM) were tested. *T. boudieri* mycelial cylinders grown on induction plates (agar-Fontana with acetosyringone) were infected with *A. tumefaciens* by dripping 30 µL of *Agrobacterium* suspension and co-cultivating for 5, 10, 30, 45, and 60 min. A sterile Whatman No. 1 filter paper disc (1–2 mm size) was used to spread the *Agrobacterium* droplet all over the mycelium during the transfer time and to remove excess cells. We considered transfer time as the duration of the droplet containing bacteria stays on the mycelium cylinder. After the removal of the filters, the infected fungal plates were incubated in the dark at 4 °C for 12 h and then cultured at 24 °C for 2, 3, 4, or 5 days in the dark. Next, to avoid further *Agrobacterium* proliferation and contamination of the cultures, the mycelial cylinders were transferred to Fontana plates containing cefotaxime (500 mg/L) and grown for 3 days in the dark at 24 °C.

### 2.5. Selection of Fungal Transformants

*T. boudieri* putative transformants were selected by growing them on Fontana-agar plates containing 25 mg/L of either geneticin or hygromycin B in the dark at 24 °C. After 2 weeks, the surviving/resistant transformants were sub-cultured on plates supplemented with an increased antibiotics concentration (50 mg/L), and peripheral growing hyphae were excised and sub-cultured under the same conditions. Selective sub-culturing was repeated three times until axenic cultures of the transformants were obtained.

### 2.6. Microscopic Analysis

Putative transformants and non-transformed control cultures were grown for 2 weeks on cellophane sheet layered on Fontana plates. The overlying mycelium was scraped and mounted on a microscope glass slide for viewing under a fluorescence microscope to identify GFP- or RFP-expressing clones (Zeiss, Jena, Germany).

### 2.7. Fungal Genomic DNA Extraction and PCR Analysis

For the genomic DNA extraction, putatively transformed mycelia cylinders were placed on a cellophane sheet layered on Fontana agar plates and grown for 2 weeks. The mycelium was scraped from the cellophane and gDNA was extracted using the DNeasy Plant Mini Kit (Qiagen, Düsseldorf, Germany) according to the manufacturer’s procedure. A touchdown PCR analysis of the GFP and RFP genes was performed using 100 ng of DNA with gene-specific primers. For the pFPL-Rh plasmid, MCHERRY-forward 5′-CCCCGTAATGCAGAAGAAGA-3′ and MCHERRY-reverse 5′-CAGGAAACAGCTATGAC-3′ were used; for the pFPL-Gg plasmid, CCDBF-forward 5′-GCATGATGACCACCGATATG-3′ and CCDBF-reverse 5′-TGTAAAACGACGGCCAGT-3′ were used (Sigma Aldrich, St. Louis, Missouri, USA). The amplification conditions included initial denaturation at 95 °C for 5 min, denaturation at 95 °C for 30 s, annealing at 56 °C for 30 s, elongation at 72 °C for 45 s with 0.2 °C decrease for each of the 30 amplification cycles, and a final elongation at 72 °C for 5 min [45]. PCR products of 1060 bp and 1801 bp for GFP and RFP, respectively, were analyzed on 1% agarose gel stained with RedSafe™ (iNtRON Biotechnology, Sagimakgol-ro, Korea).

The overall transformation efficiency was assessed taking into consideration three parameters: the number of positive resistant clones developed on selection media, fluorescence, and PCR.

### 2.8. Fungal Transgene Insertion and Stability

The transgene stability was tested by sub-culturing mycelial cylinders taken from the growing tips of a 3-week-old geneticin- and/or hygromycin-resistant cultures onto Fontana agar plates without antibiotics (25 independent events for each plasmid). Once the mycelium was spread over the plate (~2-week), a cylinder was transferred again to an antibiotics-free Fontana agar plate for up to five rounds of sub-culturing, followed by screening on a plate supplemented with 50 mg/L of geneticin or hygromycin. The putative transformed, antibiotics-resistant *T. boudieri* clones were analyzed by PCR and fluorescence microscopy. The gene-specific primers for the pFPL-Gg plasmid, EGFPN-forward 5′-CGTCGCCGTCCAGCTCGACCAG-3′ and EGFPN-reverse 5′-CATGGTCCTGCTGGAGTTCGTG-3′, and for the pFPL-Rh plasmid DSRED1C-forward 5′-AGCTGGACATCACCTCCCACAACG-3′ and DSRED1C-reverse 5′-CATGGTCCTGCTGGAGTTCGTG-3′ amplified 630 bp and 1026 bp fragments, respectively (Sigma-Aldrich, St. Louis, MO, USA). The PCR conditions and analyses were the same as described above.

### 2.9. Inoculation of the Host Plant with Transformed Hyphae

GFP- and RFP-transformed and non-transformed control mycelia were grown for 2 weeks in sterile glass tubes containing M media [46]. Then, two-day-old *H. sessiliflorum* seedlings (~1 cm long) were transferred into the tubes and co-cultivated with the fungus for 3 months at 24 °C with a 16/8 h light/dark cycle. Next, roots were collected and observed for fluorescence.

### 2.10. Data Collection and Statistical Analysis

All the experiments were performed in three biological replicates and repeated at least 3 times. Significant differences were tested by one-way ANOVA. A *p* value of <0.05 was considered as a statistically significant difference.

## 3. Results

The present research illustrates the genetic transformation of *T. boudieri*. In general, we developed an efficient *Agrobacterium*-mediated genetic transformation protocol in *T. boudieri* for future functional genomic studies. This protocol was optimized at four key stages, as follows: (1) determining the selection conditions and optimization of acetosyringone concentrations during the pre-incubation period; (2) optimizing the transfer time with *Agrobacterium* cells; (3) optimizing the mycelia developmental stage and selection with antibiotics; (4) optimizing the screening and molecular analysis of putative transformants (Appendix A). We used the protocol published by Pardo et al. and varied one of the variables while keeping the rest fixed [28].

### 3.1. Sensitivity of T. boudieri Mycelium to Antibiotics and the Effect of Acetosyringone on Transformation Efficiency

To determine the optimal concentration for antibiotic selection, mycelia cylinders were incubated on plates containing increasing concentrations of hygromycin or geneticin according to the resistance conferred by the plasmid. The sensitivity assay revealed that concentrations as low as 5 mg/L hygromycin or 15 mg/L geneticin severely restricted the growth of non-transformed cultures (Appendix A). The rate of mycelial growth declined considerably with the increase in concentrations of antibiotics, and the growth was less than 5 mm already at 2.5 mg/L hygromycin and 15 mg/L for geneticin after 2 weeks of incubation (Appendix A). In comparison, the average growth rate of mycelia in the control plates was 15 mm/2 weeks (Appendix A).

The effect of pre-incubation with acetosyringone on the efficiency of *T. boudieri* transformation was determined by testing it at 100 and 200 µM concentrations [28]. The addition of acetosyringone to the fungal plates improved the transformation efficiency in a dose-dependent manner (Figure 1). Without acetosyringone, the transformation rate was nearly zero, probably due to the low virulence of the bacterium. However, at 100 µM the transformation rate was 15% and at 200 µM it increased to 24.3% (Figure 1). Acetosyringone concentrations above 200 µM did not increase the transformation frequency.

### 3.2. Effects of A. tumefaciens Cell Density and Infection Time on T. boudieri Transformation

The optimization of *A. tumefaciens* cell density was carried out by co-cultivating mycelia cylinders with bacterial suspensions of different optical densities. The transformation efficiency was highest for the bacterial suspension of 0.3 OD_600_, with 36.7% positive transformation events (Figure 2). For suspensions with optical densities above or below 0.3, the transformation efficiency was significantly lower, indicating that this is the optimal number of cells (Figure 2A).

The optimal infection time of *T. boudieri* mycelia with *A. tumefaciens* suspension was 45 min, with a highest transformation efficiency of 37.3% (Figure 2B). Transfer times of 5, 10, and 30 min showed much lower transformation efficiencies of 1%, 3.6%, and 17.6%, respectively. Extending the incubation time to 60 min resulted in a sharp decrease in the transformation efficiency, possibly due to the over-virulence of the bacterium (Figure 2B).

### 3.3. The Effect of Pre-Selection Cultivation Time on Transformation Efficiency

Further improvement in the transformation efficiency of *T. boudieri* was accomplished by determining the optimal time of cultivation of the fungus prior to the *Agrobacteria* counter selection by cefotaxime. *T. boudieri* mycelia grown on Fontana agar plates for 3 days of exposure to *Agrobacterium* cells showed the highest transformation efficiency of 39.3% (Figure 3). Fungal cultivation for 2 days resulted in a lower transformation efficiency of about 12%. Increasing the cultivation period to 4 and 5 days significantly decreased the transformation frequency (Figure 3). When the cultivation period was extended beyond 4 days, an excessive growth of *Agrobacterium* was noted, resulting in a reduced transformation efficiency and plate contamination.

### 3.4. The Effect of the Mycelia Developmental Stage on Transformation Efficiency

Mycelia sectors along the growth axis differ in their physiological and developmental stage. It is conceivable that mycelia at various developmental stages would differ in transformation efficiency due to their distinctive cell wall structures, recovery pace, etc. Therefore, the efficiency of transformation was tested as a function of the mycelia distance from the point where growth started, along the radius from the explant primary center (marked with an arrow, Figure 4A). The excised mycelia cylinders from the middle region of the growing hyphae (1.5 to 3 cm from center, denoted as M in Figure 4A) were very dense and showed the highest frequency of transformation (17.6%). The interior zone, with the oldest hyphae closest to the center (denoted as I in Figure 4A), followed, with a transformation efficiency of 7.3%. The peripheral hyphae zone (denoted as P, Figure 4A) showed the lowest transformation efficiency of 5.3% (Figure 4B).

### 3.5. Molecular Verification of Putative Transformants

We first used the refined procedure to transform *T. boudieri* with the pFPL-Gg plasmid conferring geneticin resistance and GFP. As expected, a comparable transformation efficiency of 42% was obtained for the pFPL-Rh plasmid that has a similar backbone.

The presence of RFP and GFP transgenes in antibiotics-resistant mycelia was confirmed by PCR amplification using gene-specific primers. The expected PCR-amplified products of 1060 bp and 1801 bp for GFP and RFP, respectively, were obtained in all the selection-resistant mycelia samples (lanes 1–15 in Figure 5a,b, respectively). Only three percent false positives were observed in the PCR, however a more stringent antibiotics selection effectively eliminated them.

### 3.6. Genetic Stability of T. boudieri Transformants

To test the genetic stability of the integrated plasmid in transformed mycelia, PCR-positive clones were grown for five rounds of sub-culturing on antibiotics-free Fontana agar medium, followed by re-selection on an antibiotics-containing medium. After re-selection with antibiotics, gDNA was extracted from mycelia and analyzed by PCR. Most of the transformed clones were PCR-positive, with amplified bands of 630 bp and 1026 bp for GFP and RFP, respectively (Figure 6a,b, respectively). Out of 25 samples each for the GFP and RFP constructs, 24 and 25 samples, respectively (i.e., ~96% and 100%), were successfully amplified. This further confirmed the successful integration of the T-DNA into the fungus genome and indicated mitotic stability.

### 3.7. Expression of Reporter Genes in Free-Living Mycelia and in Mycorrhizal Association

Positive transformed mycelia, verified as described above, were also observed under a fluorescence microscope for the expression of GFP or RFP reporter genes. In most samples, the GFP-expressing mycelia had a higher fluorescence level than the RFP-harboring transformants. However, green and red fluorescence was uniformly distributed and observed mainly at the mycelial tips and septa (Figure 7b and Figure 8b). No fluorescence was detected in non-transformed mycelia except for some auto-fluorescence. The diffuse background of the auto-fluorescence observed in mock-infected mycelia (Figure 7d and Figure 8d) is in contrast with the fairly strong and localized fluorescence observed in the GFP- and RFP-expressing mycelia (Figure 7b and Figure 8b).

*H. sessiliflorum* plants were inoculated, and after three months growth with transformed mycelia the mycorrhizal association was examined for the reporter gene expression. The GFP and RFP (Figure 9 and Figure 10) expression was clearly visible in ectomycorrhizal association with host roots. The transformed mycelia showed a bright glowing fluorescence organized as glowing continuous strings in the apoplast where the hyphae wrap the cells (Figure 9b and Figure 10b), while the non-transformed mycelia showed only a uniform background auto-fluorescence (Figure 9d and Figure 10d). We could not observe glowing mycelia between the cells in controls. Significant differences were seen in the fluorescence patterns between mycorrhizal and non-mycorrhizal root cultures. In mycorrhiza, the fluorescence was uniformly distributed within hyphae that wrapped the host cells, as expected in ectomycorrhiza (Figure 9b and Figure 10b).

## 4. Discussion

The genetic intractability of desert truffles imposed a major constraint on their molecular study, and up to now no genetic transformation method has been reported for desert truffles. Since the first publication of a protocol for the genetic transformation of the filamentous fungus *Neurospora* [47], the number of reported protocols has increased rapidly [48] and they have been critically reviewed [49]. One of the primary approaches for the genetic transformation of fungi is the use of protoplasts, a technique that has been productive for a large number of fungal species [50,51,52]. However, this method of particle-mediated gene transfer [53] is tedious and time- and labor-consuming. On the other hand, *Agrobacterium*-mediated transformation was shown to induce a significantly higher transformation frequency, with more stable transformants and a lower rate of complicated insertional mutants.

Here, an efficient stable *Agrobacterium*-mediated genetic transformation system with a high frequency of DNA integration was developed and optimized for *T. boudieri*. The success of *A. tumefaciens*-mediated transformation was influenced by various factors, including the use of DNA-transfer enhancer, incubation periods, and transformant selection by antibiotics.

The phenolic acetosyringone has a prominent role in increasing virulence-gene activity which in turn triggers infection and transfer of the T-DNA into host cells [54]. The addition of this reagent into the culture medium was found to be crucial for the transformation of *T. boudieri*. Pre-growing the mycelia agar-cylinders on a medium supplemented with acetosyringone for 3 days stimulated *Agrobacterium* virulence and increased the transformation efficiency in a dose-dependent manner, with 200 μM showing the highest effect (Figure 1). In the truffle *T. borchii*, pre-growing the mycelia on potato-dextrose agar medium supplemented with 200 μM of acetosyringone also resulted in the higher transformation frequency [23,27]. Similarly, the same acetosyringone concentration was found to be optimal in several previous reports, including the ectomycorrhizal fungi *S*. *bovinus*, *H. cylindrosporum*, and *P. involutus* [28]; *S*. *bovinus* [29]; *H. cylindrosporum* [30]; *L. bicolor* [35]; and *L. laccata* [38].

Other important factors for effective transformation were the *Agrobacterium* cell density and transfer time. A cell density of 0.2 OD_600_ was reported to be the optimal for the infection of *T. borchii* with the *Agrobacterium* AGL1 strain [27] and for the transfection of *S*. *bovinus*, *H. cylindrosporum*, and *P. involutus* with the LBA1100 strain [28]. Combier et al. [26] reported that an OD of 0.15 was optimal for *H. cylindrosporum* spore transformation with AGL1. In *T. borchii*, an OD of 0.3 was reported to be optimum for both GV3101 and AGL1 strains [23]. However, for transformation of *L. bicolor* using the LBA1100 and AGL1 strains, at OD of 0.4–0.5 was optimal and an OD above 0.5 caused excessive bacterial growth and difficulties in selecting *Agrobacteria*-free transformants [35]. Our results show that mycelia of *T. boudieri* infected with *Agrobacterium* at an OD of 0.3 for 45 min enhanced the transformation efficiency by up to 40% (Figure 3). During cultivation following infection, despite the removal of excess *Agrobacterium*, some bacteria remain and their growth continues. Hence, the duration of mycelia cultivation while exposed to *Agrobacteria* must be optimized to assure efficient infection. The transformation efficiency was highest when the bacterial exposure lasted 3 days, and when extended to 4 or 5 days it decreased significantly (Figure 3). This phenomenon was probably caused by the overgrowth of *Agrobacteria*, which may have inhibited mycelia development.

The overgrowth of *Agrobacterium* cells creates a problem during selection since it interferes with fungal development and growth and may even lead to death of the mycelia. Most of the reported studies utilize single or multiple antibiotics in the selection media to control the overgrowth of *Agrobacterium*. The growth of the *Agrobacterium* strains LBA1100 and AGL1 [33,34] was controlled by the combination of cefotaxime, ampicillin, and tetracycline, each at a concentration of 100 mg/L. For the truffle *T. borchii*, 182 mg/L cefotaxime was used to eliminate the AGL1 and GV3101 strains [23]. However, in our study such concentrations of antibiotics were insufficient to control the overgrowth of strain AGL1, and a higher concentration of 500 mg/L cefotaxime was used in two selection cycles. These conditions were necessary to avoid bacterial overgrowth and to improve the transformant rescue.

A robust screening process is essential for the selection of stably transformed mycelia. A limited number of antibiotics has been reported enabling the selection of ectomycorrhizal fungi. The most important antibiotic is hygromycin for *S. bovinus* [29], *H. cylindrosporum* [25,30,31], *T. borchii* [23,27], *L. laccata* [38], and *L. bicolor* [31,32,33,34,35,36,37]. In these studies, the minimal inhibitory concentration for hygromycin ranged between 15 and 250 mg/L. Interestingly, *T. boudieri* was found to be much more sensitive to hygromycin, even at a low concentration such as 5 µg/mL and geneticin at 15 µg/mL (Appendix A). In a few mycorrhizas, geneticin was used for selection, however the minimal inhibitory concentration was 25 µg/mL, which is higher than that of hygromycin. In this study, to avoid the escape of *T. boudieri* non-transformants from selection, the cultures were subjected to a stepwise increase in the antibiotic concentration. Nevertheless, the development of non-transformed mycelia in either antibiotic selection protocol was infrequent.

The developmental stage of the mycelium was also shown to affect transformation efficiency. Mycelium excised from the middle region of the growth zone showed the highest transformation efficiency compared to the youngest peripheral mycelium or the older mycelium from the inner part of the plate (Figure 4B). Mycelia from the middle growth region are denser than the mycelia from the other parts of the growth zone. The peripheral growth zone contains hyphal tips that are more susceptible to the aggressive bacteria, probably because the cell walls are not fully established yet. It is also conceivable that, in the older growth zone, the hyphae have lower regeneration capabilities and that the thick organized cell wall structure hinders penetration.

The interaction between the plant and the mycorrhizal truffle is complex, and further research is needed to elucidate these relationships. The inoculation of *T. boudieri* with a host plant leads to better plant development, yet the mechanism by which the mycelium alters the host plant physiology is still not understood [55]. The advancement of high-efficiency molecular tools should provide in-depth knowledge of the molecular dialog between fungi and their host plants [56]. Therefore, a reliable method for genetic manipulation should be very useful in such an endeavor. In the present study, the developed method yielded ~42% transformants of *T. boudieri*, with a mitotic stability of over 90%. These rates are comparatively high for truffles. The transformants were used to inoculate *Helianthemum* roots grown under specific conditions supporting ectomycorrhiza. Indeed, the florescence of the expressed transgenes GFP and RFP was evident in the host plant apoplast (Figure 9 and Figure 10). After several decades of biochemical, physiological, and anatomical characterization of ectomycorrhiza, this molecular level study has brought novel insights [7,56]. This stable transformation system can be of aid in the future for further understanding the ability of the fungus to survive under ectomycorrhiza or endomycorrhiza lifestyles.

## 5. Conclusions

There are several possible combinations of transformation methods and functional genomics approaches, not all applicable for desert truffles. This study provides a reproducible and stable genetic transformation system which could be advantageous for improving biotechnological applications and paving the way for further in-depth research.

## Figures and Tables

**Figure 1 genes-11-01293-f001:**
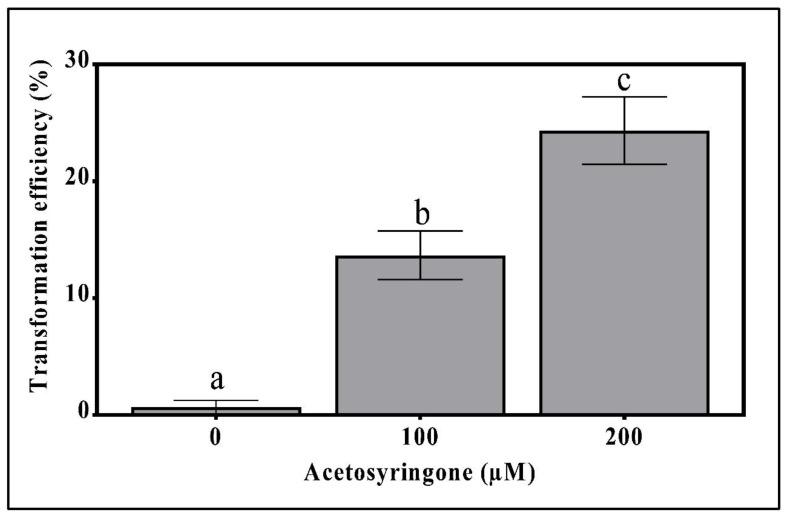
Effect of acetosyringone pre-incubation on the *T. boudieri* transformation efficiency. Mycelia containing agar cylinders were inoculated on Fontana selection media. *Agrobacterium* cells were incubated with 100 or 200 µM acetosyringone and grown to a 0.3 optical density, then incubated for a 45 min transfer time. For each treatment, 100 mycelia cylinders were infected with Agro cells. Means followed by the alphabets differ significantly at *p* = 0.05.

**Figure 2 genes-11-01293-f002:**
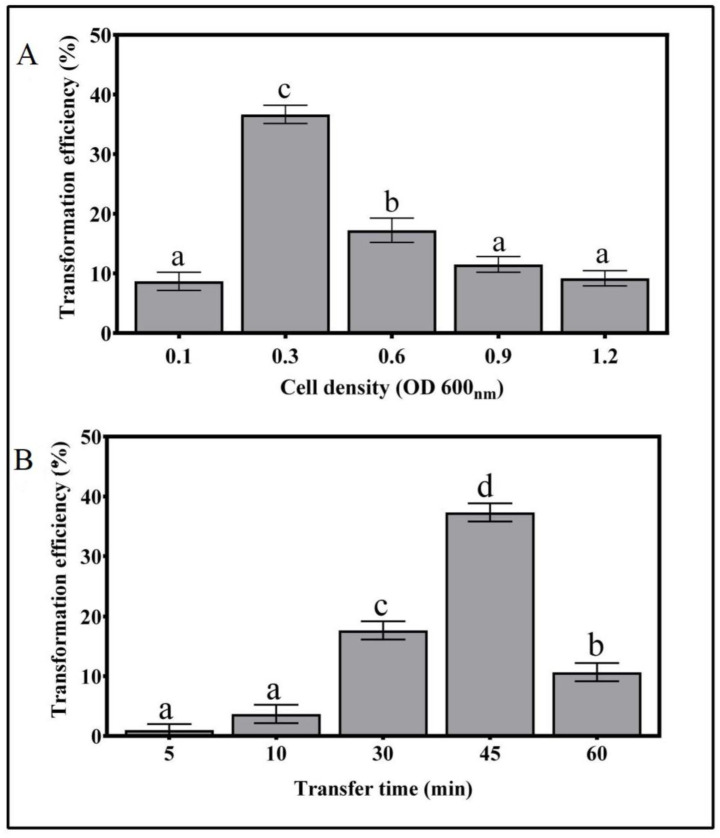
Effects of *A. tumefaciens* cell density (**A**) and transfer time (**B**) on the *T. boudieri* transformation efficiency. For each treatment 100 mycelia cylinders were infected with *Agrobacterium*. Values are mean ± SE (*n* = 3), and letters denote significant differences (*p* = 0.05).

**Figure 3 genes-11-01293-f003:**
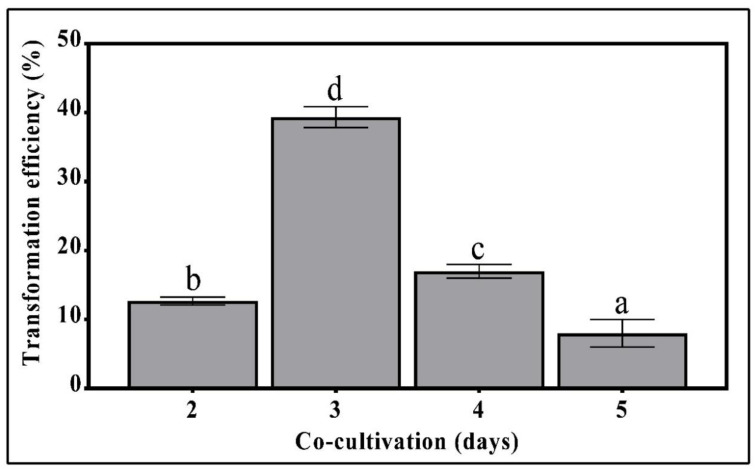
The effect of co-cultivation time prior to cefotaxime counter selection on the *T. boudieri* transformation efficiency. For each treatment, 100 mycelial cylinders were used. Values are mean ± SE (*n* = 3), letters denote significant differences (*p* < 0.05).

**Figure 4 genes-11-01293-f004:**
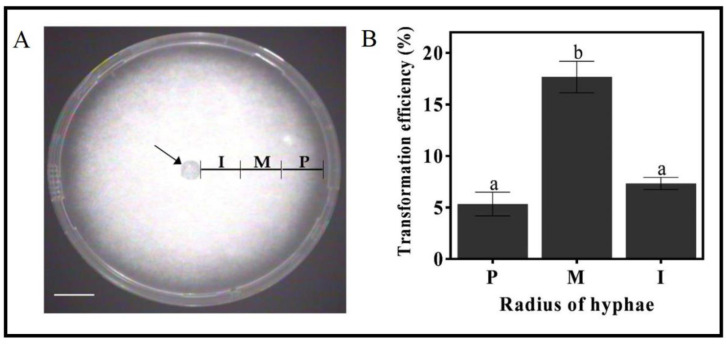
The effect of mycelia developmental stage on the *T. boudieri* transformation efficiency. (**A**) Mycelia cylinders were excised along the radius of growth. I, Interior 1–1.5 cm from the origin of growth; M, Middle 1.5–3 cm; P, Peripheral 3–4.5 cm. The plates were grown for 3 weeks on Fontana medium prior to the cylinders’ excision. Bar = 1.5 cm. (**B**) Transformation efficiency of mycelia at different developmental stages P, M, and I. Arrow denotes origin of growth. Values are mean ± SE (*n* = 3) *p* = 0.05 using Duncan’s multiple range test.

**Figure 5 genes-11-01293-f005:**
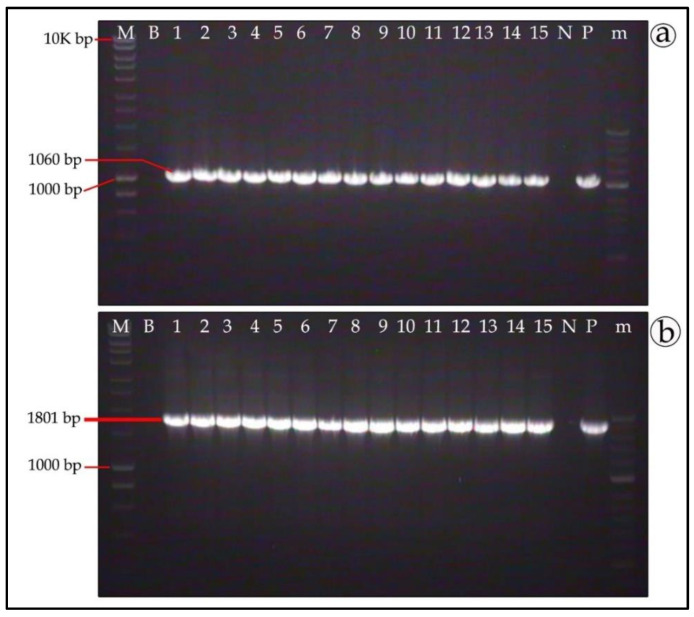
PCR verification of the *T. boudieri* transformed clones. Genomic DNA was extracted from the antibiotics-resistant mycelia of *T. boudieri* transformants. PCR amplification of green fluorescent protein (**a**) and red fluorescent protein (**b**) yielded fragments of 1060 bp and 1801 bp, respectively. M—1 kb DNA ladder; B—blank (control sample, without DNA); lanes 1—15 gDNA of various clones; N—Negative control (DNA of non-transformed mycelia); P—Positive control (plasmid); m—50 bp DNA ladder.

**Figure 6 genes-11-01293-f006:**
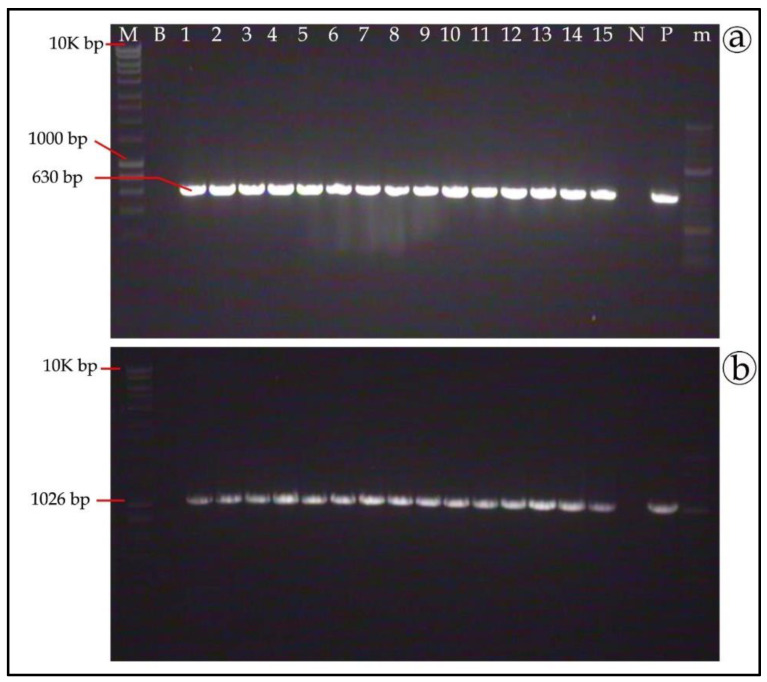
Genetic stability of the integrated plasmid in *T. boudieri* transformed clones. After 5 rounds of growth on antibiotics-free medium, gDNA was analyzed by PCR amplification for the integrated reporter genes. (**a**) Amplification of the GFP gene (amplicon size 630 bp) in gDNA extracted from geneticin-resistant mycelia. (**b**) Amplification of the RFP gene (amplicon size 1026 bp) in gDNA extracted from hygromycin-resistant mycelia. M—1 kb DNA ladder; B—blank (water control); lanes numbered 1—15 transformants gDNA amplification; N—Negative control (DNA of non-transformed mycelia); P—Positive control (plasmid); m—50 bp DNA ladder.

**Figure 7 genes-11-01293-f007:**
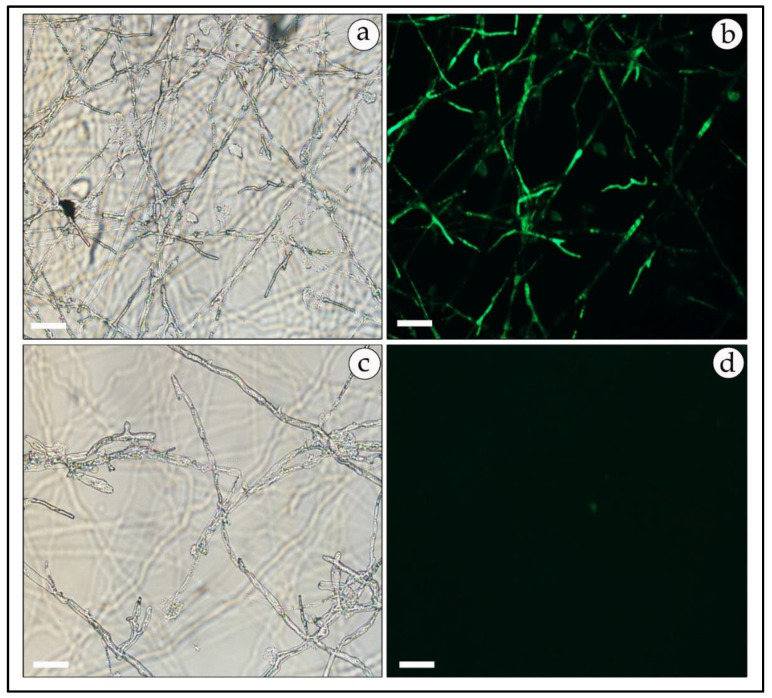
Fluorescent micrograph of GFP-expressing, geneticin-resistant mycelia of *T. boudieri*. Transformed mycelia were observed under florescent microscopically. Left panels: bright field; right panels: fluorescence microscopy. Panels (**a**,**b**)—mycelia of GFP-expressing transformants; panels (**c**,**d**)—non-transformed mycelia. 200× magnification. Bar = 20 μm.

**Figure 8 genes-11-01293-f008:**
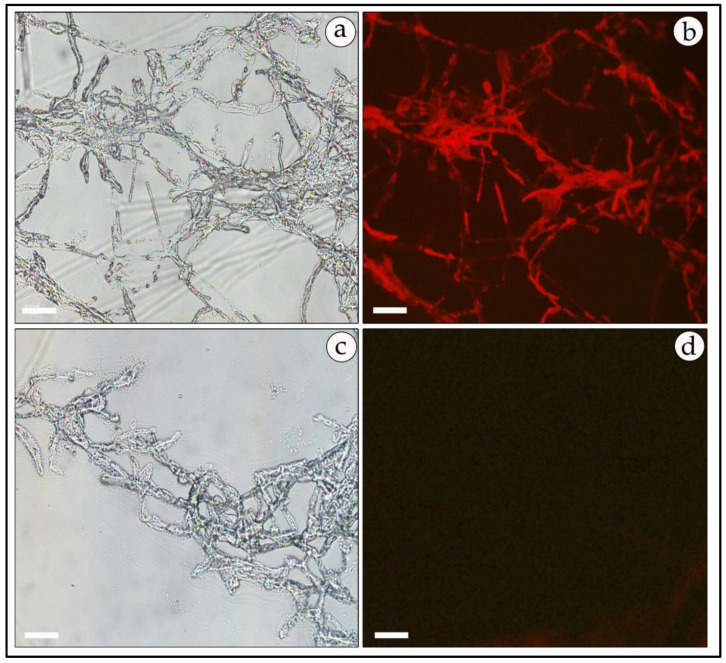
Fluorescent micrograph of RFP-expressing, hygromycin-resistant mycelia of *T. boudieri*. Transformed mycelia were observed under florescent microscopy. Left panels: bright field optics; right panels t: fluorescence microscopy. Panels (**a**,**b**)—mycelia of RFP-expressing transformants; panels (**c**,**d**)—non-transformed mycelia. 200× magnification. Bar = 20 μm.

**Figure 9 genes-11-01293-f009:**
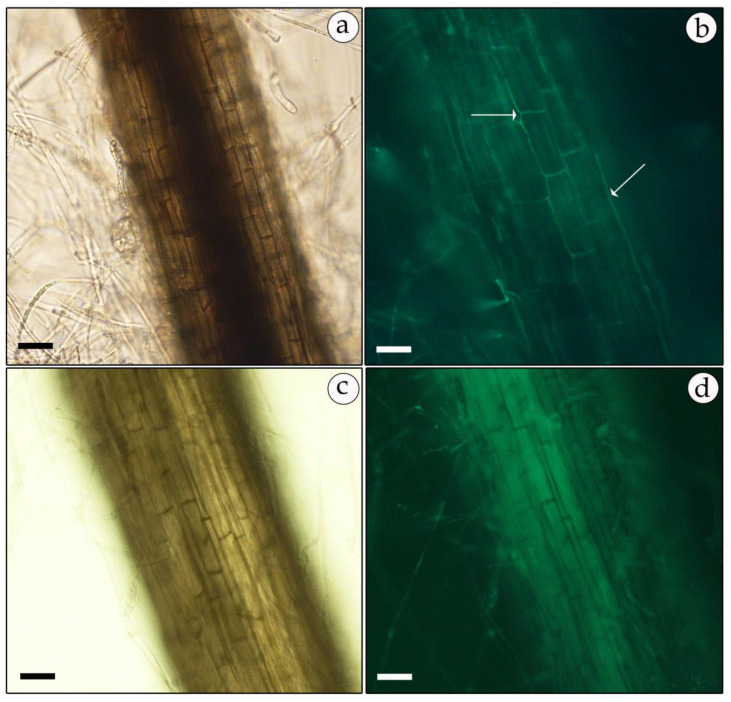
Expression of GFP in *T. boudieri* hyphae under ectomycorrhizal association with *H. sessiliflorum* roots. Left panels: bright field showing hyphae wrapping the roots (**a**). Right panel: GFP expression in the apoplast of the transformed hyphae under mycorrhiza marked with the white arrow (**b**). Non-transformed mycorrhized hyphae exhibit diffuse fluorescence background (**c**,**d**). 200× magnification. Bar = 20 μm.

**Figure 10 genes-11-01293-f010:**
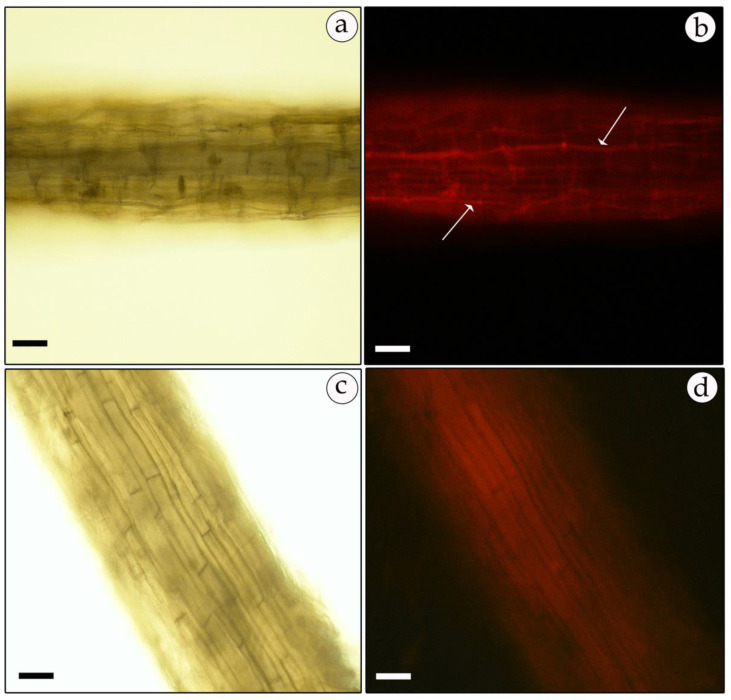
Expression of RFP in *T. boudieri* hyphae under ectomycorrhizal association with *H. sessiliflorum* roots. Left panel: bright field of mycorrhized roots (**a**). Right panel: RFP expression in the apoplast of the transformed mycelia under mycorrhiza marked with the white arrow (**b**). Non-transformed mycorrhized mycelia exhibiting diffuse fluorescence background (**c**,**d**). 200× magnification. Bar = 20 μm.

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
