# Peer review of "Agrobacterium tumefaciens-Mediated Genetic Transformation of the Ect-endomycorrhizal Fungus Terfezia boudieri"

_genes, 2020, doi:10.3390/genes11111293_

Round 1

Reviewer 1 Report

The authors developed and optimized Agrobacterium-mediated transformation protocol for a desert truffle, T. boudieri.  They were systematic in their approach and were able to reach high transformation efficiencies (40%) with mitotically-stable integration of genes that produce fluorescent proteins and antibiotic resistance enzymes. The paper is the first example of genetic modification of desert truffles and this tool/model organism will help advance the study of ect-endomycorrhizal fungi.  I recommend publication with minor modifications. 

Major comments: 

** Lines 80-82:  Can the authors be more specific on the failures of previous A. tumefaciens methods in ectomycorrhizal fungi  [Ref. 19]?  This is necessary because it highlights the significance of the authors’ results, i.e. high transformation frequency and mitotically stable transformants. The average reader is not going to read Ref. 19, so this is important. 

** It was unclear (from Materials & Methods and Fig. 1-4) how the authors optimized various stages of transformation.  Did they choose one protocol with fixed variables (concentration of acetosyringone, infection time, co-cultivation time, antibiotic counterselection of A. tumefaciens, selection of fungal transformants using geneticin / hygromycin, etc..) and systematically varied one of the variables while keeping the rest fixed. Please clarify.  Lines 186-191 might be a good place to give such an overview:  

** Figures 5 and 6 only show positive PCR signals (15/15). These figures would be more informative if they showed all tested transformants, supporting assertions that (1) “Only three percent false positives were observed in the PCR, however a more stringent antibiotics selection effectively eliminated them.” (Lines 268-269) and (2) “Out of 25 samples, each for the GFP and RFP constructs, 24 and 25 samples, respectively were successfully amplified.  This further confirmed the successful integration of the T-DNA into the fungus genome and indicated mitotic stability.” (Lines 282-284).  

 ** I could not see an obvious difference between the fluorescence images of transformed mycorrhizal hyphae and non-transformed mycorrhizal hyphae of Figs. 9 and 10.  Perhaps the authors can highlight the key differences and make it obvious to the non-expert?  

Minor comments: 

** Lines 73-75:  In addition to Ascomycetes, Basidiomycetes, and Zygomycetes, A. tumefaciens transformation has been successfully used in basal fungi, such as Chytridiomycota [ref: Medina, Robinson et al, eLife 2020]. 

** Is Figure S1 showing a flow chart of the “best” transformation protocol, or the protocol with fixed variables which were optimized one at a time (see major comment above)?  Please add missing details such as acetosyringone concentration, Agrobacterium density, developmental stage of mycelial agar cubes (middle of the plate), etc.. 

** I’m confused by the meaning of “infection time” in Figure 2b.  As far as I can tell, it is the time that a filter disc inoculated with A. tumefaciens is in contact with the mycelium.  During this time, A. tumefaciens is transferred to a mycelial plate with constant conc. of acetosyringone (which induces bacterial virulence machinery and T-DNA transfer into fungal genome). The T-DNA transfer or infection likely happens after the filter disc is removed from the mycelium. Thus, shouldn’t the “infection time” be called the “transfer time”?   

Author Response

Dear reviewer

Reviewer 2 Report

The manuscript presents a protocol for the transformation of the fungus Terfezia boudieri that produces in some arid countries an important income. Cultivation of this fungus is nowadays possible in some country, although not yet on a large scale, therefore a protocol such this presented in this manuscript can surely help in improving the procedures for increasing its production. The manuscript is clear and well written and the bibliography updated. Although this, some points need to be clarified.

Line 46: “whether” is not right

Line 62-63: not “very limited”, I suggest: For these reasons research has been carried out on desert truffle cultivation, biology and genetics (add some recent references such as Radhouani F. et al (2019) on Plant Biosystems 153: 19-24;  JE Marqués-Gálvez et al. (2019) PloS one 14 ), and the development of these valued fungi as agricultural crops is in high demand (I suggest the book Morte A. et al. (2020) or the paper Andrino et al. (2019) on  Agronomy for Sustainable Development 39)

Line 85: “enable” the study of…

Line 151: gDNA is superfluous if not compared to nuclear or mitochondrial DNA…just DNA is sufficient

Line 309: since the fungus can produce ecto and endomycorrhizae how Authors distinguished between these two different lyfestyles? This should be clearly explained

Line 355: “including the ectomycorrhizal..”

Author Response

Dear Reviewer
